# AUTOMATING THOUGHT OF SEARCH: A JOURNEY TOWARDS SOUNDNESS AND COMPLETENESS

## ABSTRACT

Large language models (LLMs) are being used to solve planning problems that require search. Most of the literature uses LLMs as world models to define the search space, forgoing soundness for the sake of flexibility. A recent work, Thought of Search (ToS), proposed defining the search space with code, having LLMs produce that code. ToS requires a *human in the loop*, collaboratively producing a sound successor function and goal test. The result, however, is worth the effort: all the tested datasets were solved with 100% accuracy. Consequently, there is great potential to automate the ToS process. We take a first major step towards automating ToS (AutoToS), taking the *human out of the loop* of interactions with the language model. AutoToS guides the language model step by step towards the generation of sound and complete search components, through feedback from both generic and domain specific unit tests. We show that AutoToS is able to achieve 100% accuracy on all the evaluated domains with a small number of LLM calls.

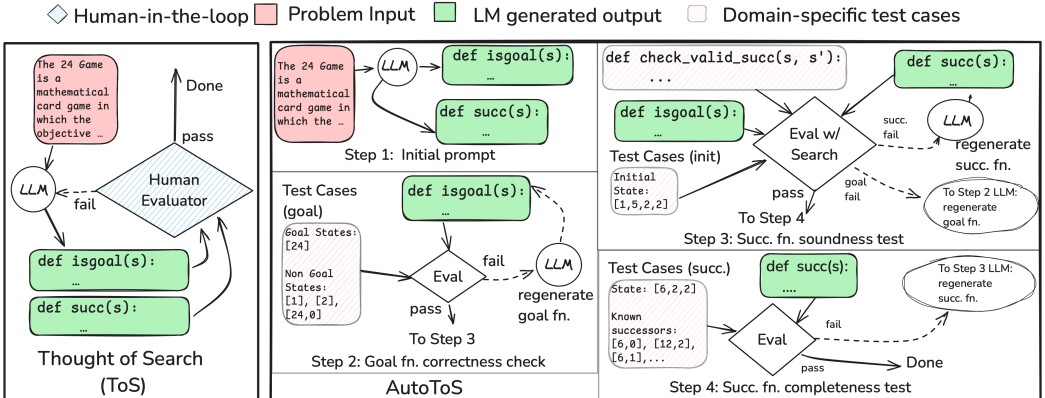

Figure 1: An overview of ToS and AutoToS.

## INTRODUCTION

Large language models (LLMs) have shown great promise across countless domains and fields, especially as their architectures become more advanced. Spurred by their abilities in natural language tasks, recently LLMs were used for solving AI planning tasks. The approaches vary from giving a planning problem to an LLM and asking it to output an entire plan at once (Silver et al., 2022; Kambhampati et al., 2024; Pallagani et al., 2022) to asking an LLM to generate a planning model to be given to an automated planner (Guan et al., 2023; Oswald et al., 2024; Gestrin et al., 2024). Between these two extremes, lies a body of work on using LLMs to plan by performing a combinatorial search (Hao et al., 2023; Yao et al., 2023; Besta et al., 2024; Sel et al., 2023). Among these, Thought of Search (ToS) (Katz et al., 2024) stands out; it uses LLMs to define the search space for the entire domain at once, by soliciting two crucial search components, a successor function and a goal test. These components are then plugged into a standard search algorithm, such as Breadth-First Search (BFS) or Depth-First Search (DFS) (Cormen et al., 1990). ToS has an impressive accuracy of 100% on all tested benchmarks and it produces a symbolic model whose soundness and completeness can be verified. However, ToS has a significant limitation - it requires a human expert in the loop,

iteratively providing feedback to the model on the produced code. Our contribution is precisely there. We take the first major step towards automating the ToS process, starting with the feedback loop.

We automate the iterative feedback and exception handling process through the use of unit tests and print debugging statements, incorporated with Chain of Thought (CoT) style prompting (Wei et al., 2022). We remove the human expert interaction with the LLM, shifting the human involvement to specifying unit tests. We note that unit tests often exist already for search problems or can be generated with the help of LLM (Pan et al., 2025). Our approach leverages these existing unit tests (referred to henceforth as domain-specific unit tests), along with a small number of predefined generic domain-independent unit tests. The manual involvement in creation of unit tests is therefore minimal.

We demonstrate our approach on five representative search problems of various complexity and benchmark using a variety of LLMs. We find that the accuracy of the code generated by LLMs with automated feedback consistently reaches 100% in at least one trial of most models for every tested domain, while the total number of LLM calls remains small, comparable to the results of ToS with human feedback. In an ablation study, we show the importance of soundness and completeness feedback for obtaining highly accurate final code. Finally, we investigate the errors in the code generated by LLMs and find that the tested models differ significantly in error type distribution.

## RELATED WORK

**Planning with LLMs** Valmeekam et al. (2023b) analyzed LLMs ability to generate plans for classical planning problems described in natural language. Raman et al. (2022) generated task plans and used precondition errors as feedback to revise them. External validators were used to feedback LLMs, to generate better plans (Stechly et al., 2024; Kambhampati et al., 2024). Pallagani et al. (2023) investigate training approaches to plan generation. All these approaches use LLMs to solve one problem at a time–essentially treating LLM as a policy. Another line of work extracted policies or generalized plans from LLMs. Silver et al. (2024) synthesized generalized plans as Python programs for planning domains described in a formal language (PDDL). LLMs were also used to extract planning problems and models from their natural language description, either domain model (Guan et al., 2023; Gestrin et al., 2024; Oswald et al., 2024) or instance (Liu et al., 2023; Zuo et al., 2024; Xie et al., 2023). However, LLM generated PDDL remains less reliable and difficult to evaluate.

**Planning with LLMs using Search** A flourishing research direction pairs LLMs with external search mechanism (MCTS, BFS, DFS, etc.) (Hao et al., 2023; Yao et al., 2023; Besta et al., 2024; Zhou et al., 2023; Shinn et al., 2023). The idea is to treat LLMs as world models to generate next states and as reasoning agents to guide the search. While these approaches report higher accuracy than the basic methods, albeit possibly due to memorization (Katz et al., 2025), their significant reliance on LLMs for generating successors makes them extremely inefficient and unreliable. Thought of Search (ToS) (Katz et al., 2024), on the other hand, proposed using LLMs to generate code for the successor and goal functions, capturing domain dynamics. Once these functions are available, any search algorithm can be used to solve any problem in the domain. While this approach is significantly more efficient than using LLMs in the loop during search, it requires feedback from a human expert. Our work focuses on alleviating the requirement of human in the loop feedback. Another line of work deals with using LLMs to generate code for the search guidance (Tuisov et al., 2025; Corrêa et al., 2025). Our work is complementary and in fact it provides the prerequisite for these efforts.

**Code Generation with LLMs** LLM generated code correctness is evaluated by a variety of benchmarks (Chen et al., 2021; Puri et al., 2021; Li et al., 2024). Subsequent approaches have demonstrated human level performance on coding benchmarks (Zhong et al., 2024; Muennighoff et al., 2024). Execution errors were used to feedback LLMs for code refinement (Chen et al., 2024; Zhang et al., 2023) and external verifies are used to curate the feedback (Madaan et al., 2023; Gou et al., 2024; Huang et al., 2024), as well as unit test results (Jiang et al., 2023). Inspired by them, we propose to automate the feedback of ToS with both generic and domain-specific unit tests and validators.

## BACKGROUND

In this work we follow the notation of Katz et al. (2018), slightly adapting it for our purposes. A deterministic planning problem over a state space is a tuple $\Pi = \langle S, A, s_0, S_G, f \rangle$, where $S$ is a finite

set of *states*, $A$ is a finite set of *action labels*, $s_0 \in S$ is the *initial state*, $S_G \subseteq S$ is the set of *goal states*, and $f : S \times A \to S$ is the *transition function*, such that $f(s, a)$ is the state which applying action $a$ in state $s$ leads to. A triplet $\langle s, a, f(s, a) \rangle$ is called a *transition*. A *solution* to such a problem is a sequence of states and action labels (also called a trace) $\rho = \langle s_0, a_1, s_1, a_2, \dots a_n, s_n \rangle$, such that $f(s_i, a_{i+1}) = s_{i+1}$ for $0 \leq i < n$ and $s_n \in S_G$. In cases when the action labels are not important, they can be dropped from the definition.

The "black box" approach encodes the state space with a tuple $\Pi_{bb} = \langle s_0, succ, isgoal \rangle$, where $s_0$ is the initial state, $succ : S \to 2^{A \times S}$ is a successor generator, and $isgoal : S \to \{T, F\}$ is the goal test. A *solution* to the black-box problem is a sequence of states and action labels (a trace) $\pi = \langle s_0, a_1, s_1, a_2, \dots a_n, s_n \rangle$, such that $\langle a_{i+1}, s_{i+1} \rangle \in succ(s_i)$ for $0 \leq i < n$ and $isgoal(s_n) = T$. Here as well, if action labels are not important, they can be dropped.

We now establish the correspondence between the black-box encoding and the planning problem.

**Definition 1** (Soundness and completeness)**.**
*isgoal is* **sound** *if* $\quad \forall s \notin S_G, isgoal(s) = F$ *and* **complete** *if* $\quad \forall s \in S_G, isgoal(s) = T$ .
*succ is* **sound** *if* $\quad succ(s) \subseteq \{ \langle a, f(s, a) \rangle \mid a \in A \}$ *and* **complete** *if* $\quad succ(s) \supseteq \{ \langle a, f(s, a) \rangle \mid a \in A \}$.

Sound and complete successor generator and goal test provide the "black box" description of the state space of the planning problem $\Pi$. In such cases, a solution to $\Pi_{bb}$ is guaranteed to be a solution to $\Pi$, and if no solution for $\Pi_{bb}$ exists, then $\Pi$ also must be unsolvable. If the successor generator and goal test are sound, but not necessarily complete, it is still the case that a solution to $\Pi_{bb}$ is guaranteed to be a solution to $\Pi$. Therefore, soundness allows us to reliably use $\Pi_{bb}$ for producing solutions for $\Pi$.

## PROPOSED APPROACH AND METHODOLOGY

We build on prior work that proposed producing a code implementation of $succ$ and $isgoal$ functions (Katz et al., 2024), but we seek to remove the human out of the feedback loop of evaluating the functions. Similar to that work, we care about two properties, *soundness* and *completeness*. As we deal with planning problems described in a natural language, we do not have the formally defined planning task $\Pi$. Although not stated formally, previous work on generating $succ$ and $isgoal$ with LLMs assumes the existence of a human expert with the ability to access $\Pi$ (often in their mind). Examples of such access include feedback on the code of $succ$ and $isgoal$ produced by the LLM (Katz et al., 2024) or validating a solution obtained from the LLM in cases when $succ$ and $isgoal$ are implemented through LLMs (Hao et al., 2023; Yao et al., 2023; Besta et al., 2024; Sel et al., 2023). Here, we make a similar assumption, but request a different access to $\Pi$. We assume the existence of unit tests (either produced by LLMs, or by a human expert) that provide evidence of unsoundness or incompleteness. The evidence is then used to automatically feedback the model with the specific information needed to fix the code. We deal with three types of information, as illustrated with the 24 Game (Yao et al., 2023), where one performs arithmetic operations on 4 given numbers to reach 24.

- Examples of inputs to $isgoal$ for which the correct output is known. For instance, we know that $isgoal([24])$ should be true and $isgoal([24, 1])$ should be false.
- Examples of inputs to $succ$ for which some of the correct outputs are known. For instance, we know that [24], [2], and [-2] are valid successors of [6,4] and should be in $succ([6, 4])$.
- A partial soundness check for a transition $\langle s, a, t \rangle$ quickly invalidating (obviously) incorrect transitions. For instance, in 24 Game we know that it must be that $|t| = |s| - 1$.

The first two are usually readily available and often come with the description of the problem. The third one might require some level of understanding of the problem being solved, and it is domain specific (i.e., varies with the search problem), but the framework allows the use of a trivial partial soundness test that always reports no issues. As stated earlier, the above categories of unit tests can be produced relatively easily by an LLM or a human, and are domain-specific. Figure 1 presents an overview of our approach, describing how the provided information is used.

Step 1 Following Katz et al. (2024), we ask for the successor function $succ$ and the goal test $isgoal$.

Step 2 Then, we perform the goal unit tests, providing feedback to the model in cases of failure, repeatedly asking for a new $isgoal$ until all goal unit tests have passed, or until a predefined number of iterations.

Step 3   Once *isgoal* has passed the unit tests, we perform a soundness check of the current *succ* and *isgoal* functions. We do that by plugging these functions in a BFS extended with additional checks and run it on a few example problem instances. If BFS finished, we check if the goal was indeed reached. If not, that means that *isgoal* failed to correctly identify a state as a non-goal state and we provide that as feedback to the model, repeating Steps 2 and 3.

Step 4   (Optional) Finally, we perform the successor completeness unit test, providing feedback to the language model in case of failure.

If the goal test fails, we go back to Step 2, if the successor test fails, we go back to Step 3. After Step 1, we always have *succ* and *isgoal* that can be plugged into a search algorithm. Step 3 failing is an indication that we cannot trust the solutions produced by that algorithm. Example feedback produced in Steps 2, 3, and 4 can be seen in Figure 2. In what follows, we provide detailed description of each step of AutoToS.

**System prompt** We instruct the model to provide answers in convenient form for integrating as a search component. The produced code should consist of a single, self-contained function. Following existing work (Zhong et al., 2024; Yang et al., 2024), we use the system prompt:

> You are a Python coding assistant. Help me generate my Python functions based on the task descriptions. Please always generate only a single function and keep all imports in it. If you need to define any additional functions, define them as inner functions. Do not generate examples of how to invoke the function. Please do not add any print statements outside the function. Provide the complete function and do not include any ellipsis notation.

**Step 1: Initial prompt** While the initial prompt is the primary source of information for the language model and therefore very important, we assume that we have very limited control over it. We therefore mostly take the existing initial prompt from previous work, only ensuring that it includes an example input to the requested function in the correct format (Katz et al., 2024).

**Step 2: Goal function correctness check** Goal unit tests assume the existence of a few *known* goal and non-goal states. If the goal function *isgoal* incorrectly identifies a goal state, then it is incomplete, according to Definition 1. If it incorrectly identifies a non-goal state, then it is not sound. A search with a non-sound goal function can incorrectly report that a solution was found. One illustrative example from the 24 Game is a state [24, 1], which a goal test function may incorrectly identify as a goal state and stop before the actual solution was found – in this case, another arithmetic operation was needed. Whenever an issue with either goal function soundness or completeness was identified, we give feedback to the language model with the description of the failure and the state for which the failure occurred. See Figure 2 (top) for an example feedback. Here and later, we use a chain of thought style request, asking the model to discuss why a mistake was made and to come up with a fix.

**Step 3: Successor function soundness check** A soundness check assumes the existence of example problem instances for which we know how to validate that a goal was reached. We extend the BFS/DFS search with additional checks as follows. First, both the successor and goal test functions are wrapped with a timeout of 1 second. These functions should be able to finish in a few milliseconds and therefore 1 second timeout is an indication of an issue with the function. An issue can be as simple as unnecessary computation or multiple successor steps performed instead of a single step or it can even be an infinite loop. Second, the successor function is wrapped with a check whether it modifies the input state. Such modifications often happen when successor states are copied from the input state and modified. A shallow copy of the input state was observed in the previous work (Katz et al., 2024). These two types of tests are domain-independent and baked into the search process. Third, for every successor generated at the expansion step of BFS, a partial soundness check is performed, examining the validity of transitioning from the parent state to the successor state. An example of such a partial soundness check in 24 Game is that the successor state size must be one number less than the parent state. If that does not hold, the successor function is not sound according to Definition 1. It is worth emphasizing that this partial soundness check may be more difficult for an LLM or human to generate and therefore we treat it as an optional check. If any of the checks did not pass, we provide feedback to the LLM with the respective error message, providing example input state and the unexpected (or expected and unobserved) output, until all tests are passed or a predefined number of iterations was exhausted. See Figure 2 (middle) for an example feedback.

**Step 4: Successor function completeness check** A successor function completeness check assumes the existence of a few *known* parent and successor states. These can include all successors for some parent state or a subset thereof. If the successor function does not produce some of the known

> The goal test function failed on the following input state [24, 1], incorrectly reporting it as a goal state. First think step by step what it means for a state to be a goal state in this domain. Then think through in words why the goal test function incorrectly reported input state: [24, 1] as a goal state. Now, revise the goal test function and ensure it returns false for the input state. Remember how you fixed the previous mistakes, if any. Keep the same function signature.

> Invalid transformation: length mismatch - the length of a successor must be one less than the parent. Let's think step by step. First think through in words why the successor function produced a successor that had a length that was not exactly one less than the parent. Then provide the complete Python code for the revised successor function that ensures the length of a successor is exactly one less than the parent. Remember how you fixed the previous mistakes, if any. Keep the same function signature.
> Input state: [1, 1, 4, 6] Example wrong successor state: [6, 5]

> Successor function when run on the state [1, 1, 4, 6] failed to produce all successors. Missing successors are: [[1, 4, 7], [-5, 1, 4], [1, 1, 2], [1, 5, 6], [0.25, 1, 6], [-3, 1, 6], [0.16666666666666666, 1, 4], [1, 3, 6], [1, 4, 5], [1, 1, 1.5]] First think step by step why the successor function failed to produce all successors of the state. Then, fix the successor function. Remember how you fixed the previous mistakes, if any. Keep the same function signature.

Figure 2: **24 Game** example feedback.

successors, then it is not complete according to Definition 1. While completeness is not required for producing valid (sound) solutions, incomplete functions may not generate the part of the search space where goal states are located and therefore may not be able to find solutions. Improving completeness is therefore an optional step that may improve the accuracy of the produced code. Here as well, we give feedback to the language model with the respective error message, providing example input state and the missing successors. See Figure 2 (bottom) for an example feedback.

**Automation, evaluation and validation** Since the expensive calls to large language models are not performed during search, there is no need to artificially restrict the algorithms to their incomplete variants ( e.g., Yao et al. (2023)). Sound and complete algorithms BFS/DFS can be used for solving the search problems. Still, as the human feedback is *before* the feedback loop and the search components produced are not *guaranteed* to be sound, the solutions produced must be validated for soundness.

## EXPERIMENTS

To validate the feasibility of our approach, AutoToS, we conduct experiments on a representative collection of five search/planning problems of varying computational complexity: BlocksWorld (Gupta & Nau, 1992), PrOntoQA (Saparov & He, 2023), Mini Crossword , 24 Game (Yao et al., 2023), and Sokoban (Junghanns & Schaeffer, 1997). Four of these domains appeared in ToS (Katz et al., 2024), Sokoban domain did not. Two of these domains are polynomial, two are NP-complete, and one is PSPACE-complete. We test the performance of various LLMs from 3 families, using both the largest and smallest models from the same family. Specifically, we use GPT-4o and GPT-4o-Mini (OpenAI et al., 2024), Llama3.1-70b and Llama3.1-405b (Dubey et al., 2024), as well as DeepSeek-CoderV2 (DeepSeek-AI et al., 2024). We additionally tested Llama3-70b (AI@Meta, 2024), Mistral7x-8b (Jiang et al., 2024), and DeepSeek-CoderV2-Lite, finding these models to perform poorly and hence excluded from consideration. We use greedy decoding with maximum context length for each model. For each domain, we restrict the number of calls to the language model per function to 10. We repeat each experiment 5 times.

Following ToS, we use a simple implementation of BFS and DFS search algorithms in Python. DFS is used for Mini Crosswords, while BFS is used for the other 4 domains. Each successor function execution is limited to 1 second and each overall search is limited to 600 seconds. For each domain, a few (up to 10) instances are used for creating the unit tests. In one case, these instances are taken out of the available set of instances, in other cases we invent new instances. The rest are used for evaluating the accuracy of the generated code, where accuracy measures the percentage of the instances solved. In the case of BFS search, we also require the solution produced to be optimal. This is relevant to BlocksWorld and Sokoban where the solution length matters, but irrelevant for PrOntoQA, where solution is a boolean answer, and 24 Game, where all solutions are of the same length. It is important to emphasize again that if successor function and goal test are sound and complete, then the solution produced by BFS/DFS is guaranteed to be correct (and in the case of BFS optimal). However, since no such guarantees are available, we automatically validate every solution obtained. Experiments were performed on a AMD Ryzen 7 4800H. OpenAI models were accessed via an API, while Llama and DeepSeek were interacted with through a chat user interface.[1]

---

[1]The data, unit-tests, the code, and the model correspondence logs are provided in the supplemental material.

The aim of our evaluation is to test the following hypotheses. First, whether a partial soundness test improves the accuracy of AutoToS. Second, whether the (optional) completeness step improves the accuracy of AutoToS. Third, whether the number of calls to the language model increases significantly compared to ToS. Finally, whether the performance of AutoToS is consistent across different language models of varying sizes. We thus view the models performance without any feedback as our baseline.

**24 Game** The 24 Game (Yao et al., 2023) takes 4 integers as an input that can be manipulated through the four most common arithmetic operations: addition, subtraction, multiplication, and division. The goal of the game is to reach 24, if possible. States are represented as lists of length 4 or less.

*Data:* We use the set of 1362 instances (Yao et al., 2023; Katz et al., 2024) and we take out the first 10 instances for unit tests. Goal unit tests use [24] for goal and [], [3] ,[24, 1], [1, 6, 4], [1, 1, 4, 6] for non-goal examples. Successor completeness test uses the initial state with all its successors for each of the 10 instances, as well as a single transition along a known solution path for each of these instances. For example, the successors of [6, 6, 6, 6] are [1, 6, 6], [6, 6, 12], [0, 6, 6], and [6, 6, 36]. Also, a successor of [6, 6, 12] along the known solution path is [6, 18] and of [6, 18] is [24].
*Partial soundness test:* For the partial soundness test we check whether the number of elements in a successor state is one less than for the parent state.
*Solution validation:* A solution is a sequence of states $s_0, s_1, s_2, s_3$, where $s_0$ is the initial state, $s_3 = [24]$ is the goal state, and $\langle s_0, s_1 \rangle$, $\langle s_1, s_2 \rangle$, and $\langle s_2, s_3 \rangle$, are valid transitions. We check that all these hold for a given sequence.

**BlocksWorld** BlocksWorld is a classic AI planning domain, where the task is to rearrange blocks in towers (Gupta & Nau, 1992). There are 4 actions: stack a block on top of another block, unstack a block from another block, put a block down on the table, and pick a block up from the table. States are represented as dictionaries based on 'clear', 'on-table', 'arm-empty', 'holding', and 'on', describing whether a block is clear (no block above it in the tower), the block is on the table, whether the arm is not holding a block and which blocks are on which.

*Data:* The domain has a PDDL representation and a large collection of 502 instances was created by Valmeekam et al. (2023a) and used in the recent literature (Hao et al., 2023). We use the entire collection for evaluation and invent 2 example states (and transitions along 2 plans) per unit test for the successor completeness tests.
*Partial soundness test:* For the partial soundness test we notice that in each tower there is a top block (that is clear) and there is a bottom block (that is on the table). Therefore we simply check that the number of blocks in the 'clear' list is the same as in the 'on-table' list.
*Solution validation:* As the instances are given in PDDL, we simply translate the solution into a PDDL format and use an external validator VAL (Howey & Long, 2003).

**Mini Crosswords** The mini crosswords (Yao et al., 2023) is a 5x5 crosswords dataset where the input describes the 5 horizontal and 5 vertical clues and the output is the full 25 letters board. We provide a list of horizontal and vertical clues which are strings of words. The verifier ensures that the size of each word in the rows or columns does not exceed 5.

*Data:* We use the existing 20 instances (Yao et al., 2023; Katz et al., 2024), all used for evaluation, with the unit tests constructed based on 3 invented states each to minimize use of examples for feedback, with the successor completeness based on a state in which one horizontal and one vertical clue already filled, which limits the number of possible successors considerably.
*Partial soundness test:* The partial soundness test verifies that at most 5 new letters are filled in one transition, as well as that the number of unfilled letters does not get larger.
*Solution validation:* A crossword puzzle is solved if the end result is valid, meaning every vertical and horizontal clue is present in the list of possible clues.

**PrOntoQA** Logical reasoning can be viewed as a search problem of finding a sequence of logical rules that when applied to the known facts, derive or disprove the target hypothesis. Previous work applies MCTS with successor function and rewards obtained by calling an LLM, to examples from the PrOntoQA dataset (Saparov & He, 2023) to derive the answer but also the proof, a sequence of reasoning steps. A state is therefore a set of the facts which are known to be true.

*Data:* We use the existing set of 4000 instances entirely for evaluation, inventing 3 examples per unit test as in the other search problems.
*Partial soundness test:* A partial soundness test simply checks that each transition adds a single

known fact to the state, ensuring that the state size increases by exactly 1.
*Solution validation:* In order to validate the solution, we compare to the known correct answer.

**Sokoban** Sokoban (Junghanns & Schaeffer, 1997) is a planning problem of PSPACE-complete complexity even for non-optimal planning. The problem, despite its simple conceptual rules, is notoriously hard for generic AI planners and even for specialized solvers. We use a 2-D grid setup, in which, given equal number of boxes and goal squares, the player needs to push all boxes to goal squares without crossing walls or pushing boxes into walls. The player can only move or push up, down, left and right, and some pushes are irreversible. States are represented as dictionaries with entries: 'at-player,' which represents a single pair of coordinates, and 'at-stone', a list of coordinates for the stones. The domain has a PDDL planning model.

*Data:* We use the collection of PDDL problem instances from the International Planning Competition (IPC) 2008. Out of these instances, we select a subset that can be solved relatively quickly by using the blind search configuration of the efficient planner Fast Downward (Helmert, 2006) and choose the instances that were solved in under 5 seconds. This resulted in 11 instances. We use the entire set for evaluation and invent 3 states per unit test.
*Partial soundness test:* The test checks if the locations of the player and the stones are all different.
*Solution validation:* Similar to BlocksWorld, we translate the solution to PDDL format and use VAL.

**Evaluation Accuracy** Figure 3 depicts the progression of the average accuracy values across the 5 domains and 5 trials, comparing using the partial soundness test (solid lines, '+PST') and not (dotted lines). Each color represents a language model. The first point in the process corresponds to when the search components are first created, meaning no feedback at all - our baseline. The second point is when the goal and successor function soundness tests are not failing. The third and final point is the end of the process, when successor completeness tests are not failing. The aggregation is performed over the cases when the step was reached. The figure allows us to find answers for both the first and the second hypotheses. Recall that we have access to a validator, and therefore we can take advantage of the multiple repetitions, taking the best result instead of average. Table 1 depicts the per-domain results for the best out of 5 trials accuracy, comparing across models, both with and without partial soundness test. We see that 100% accuracy is achieved for all domain, across models, with only a few exceptions.

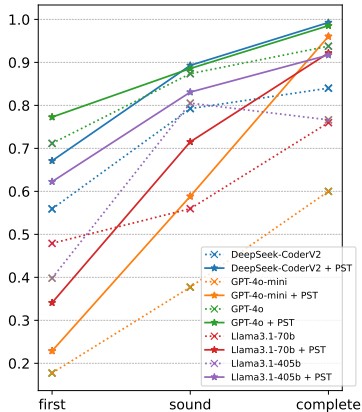

Figure 3: Progression of average accuracy values during AutoToS.

We can clearly see the benefit from using the partial soundness test, even as simple as the ones we described above. Going forward, we therefore restrict our attention to using the partial soundness test. Further, we can clearly see the strong increase in accuracy when not stopping after the soundness test passes and performing the completeness tests, across all models.

| model | 24 Game | | | BlocksWorld | | | Crossword | | | Sokoban | | | PrOntoQA | | |
|---|---|---|---|---|---|---|---|---|---|---|---|---|---|---|---|
| | F | S | C | F | S | C | F | S | C | F | S | C | F | S | C |
| DeepSeek-CoderV2 | 1.0 | 1.0 | 1.0 | 1.0 | 1.0 | 1.0 | 1.0 | 1.0 | 1.0 | 1.0 | 1.0 | 1.0 | 1.0 | 1.0 | 1.0 |
| DeepSeek-CoderV2+PST | 1.0 | 1.0 | 1.0 | 1.0 | 1.0 | 1.0 | 1.0 | 1.0 | 1.0 | 1.0 | 1.0 | 1.0 | 1.0 | 1.0 | 1.0 |
| GPT-4o-mini | 0.0 | 1.0 | 1.0 | 0.0 | 0.11 | NA | 1.0 | 1.0 | 1.0 | 0.0 | 0.0 | 0.0 | 1.0 | 1.0 | 1.0 |
| GPT-4o-mini+PST | 0.64 | 1.0 | 1.0 | 0.0 | 0.83 | 0.83 | 1.0 | 1.0 | 1.0 | 0.0 | 1.0 | 1.0 | 1.0 | 1.0 | 1.0 |
| GPT-4o | 1.0 | 1.0 | 1.0 | 1.0 | 1.0 | 1.0 | 1.0 | 1.0 | 1.0 | 1.0 | 1.0 | 1.0 | 1.0 | 1.0 | 1.0 |
| GPT-4o+PST | 1.0 | 1.0 | 1.0 | 1.0 | 1.0 | 1.0 | 1.0 | 1.0 | 1.0 | 1.0 | 1.0 | 1.0 | 1.0 | 1.0 | 1.0 |
| Llama3.1-70b | 1.0 | 1.0 | 1.0 | 0.0 | 0.37 | 1.0 | 1.0 | 1.0 | 1.0 | 0.0 | 0.0 | 0.0 | 1.0 | 1.0 | 1.0 |
| Llama3.1-70b+PST | 1.0 | 1.0 | 1.0 | 0.0 | 1.0 | 1.0 | 0.15 | 1.0 | 1.0 | 0.0 | 0.82 | 0.82 | 1.0 | 1.0 | 1.0 |
| Llama3.1-405b | 1.0 | 1.0 | 1.0 | 0.0 | 1.0 | 1.0 | 0.0 | 0.95 | NA | 1.0 | 1.0 | 1.0 | 1.0 | 1.0 | 1.0 |
| Llama3.1-405b+PST | 1.0 | 1.0 | 1.0 | 0.0 | 1.0 | 1.0 | 1.0 | 1.0 | 1.0 | 1.0 | 1.0 | 1.0 | 1.0 | 1.0 | 1.0 |

Table 1: Domain-wise best-of-5 accuracy. F/S/C stand for First, Sound, and Complete, respectively. NA: no trials passed completeness. +PST: with partial soundness test.

**Number of LLM calls** Table 2 shows the total number of calls to the language model until soundness and completeness tests pass. Note that the minimum number of calls is 2, one for each component, even without feedback. The number of automated calls is comparable to the

| | | 24 Game | PrOntoQA | Sokoban | Crossword | BlocksWorld |
|---|---|---|---|---|---|---|
| AutoToS | GPT-4o-mini | 8.8 | 4.8 | 6.4 | 9.6 | 10.0 |
| | GPT-4o | 3.4 | 2.6 | 2.2 | 5.8 | 2.0 |
| | Llama3.1-405b | 3.4 | 2.0 | 2.6 | 4.0 | 3.2 |
| | Llama3.1-70b | 7.4 | 2.0 | 8.2 | 6.2 | 5.8 |
| | DeepSeek-CoderV2 | 4.4 | 2.0 | 2.8 | 6.6 | 4.2 |
| ToS | GPT-4 | 2.2 | 2.6 | NA | 3.8 | 3.8 |

Table 2: Average per domain number of LLM calls.

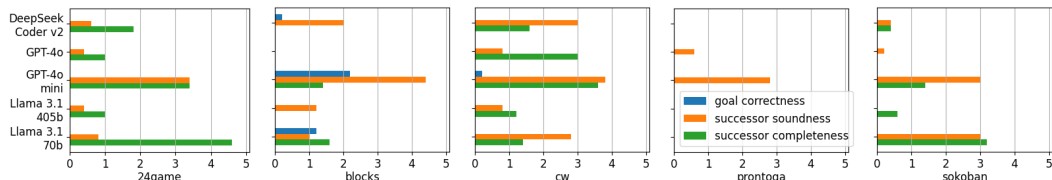

Figure 4: Average number of feedback calls for goal correctness, successor soundness/completeness.

one when a human expert is giving the feedback to the model. To look deeper into how the feedback is partitioned among the three phases, Figure 4 compares the numbers across language models and domains. We see that the larger models rarely require any feedback on the goal function and only a few iterations on the successor function, and more often than not on completeness.

Finally, we can observe that there is no single model that performs better than all other, according to all parameters and the performance is quite consistent across the large models. Interestingly, the smaller model GPT-4o-mini performs quite well in terms of accuracy.

## CODE ERRORS DISCUSSION

To be able to improve the performance of the large language models in generating search components, it is important to understand the errors in the code produced by these models. In the following section we first present the error categories and show the partitioning of the errors to these categories and then elaborate on a few interesting cases.

**Error categories** AutoToS distinguishes 10 error categories and gives each a separate feedback.

1. $succ$ soundness test failed.   2. Input state changed by $succ$.   3. $succ$ completeness failed.
4. $isgoal$ soundness failed.   5. $succ$ exception occurred.   6. $isgoal$ exception occurred.
7. Timeout in $succ$ soundness test.   8. Timeout in $succ$ execution.   9. Timeout in $isgoal$ execution.
10. Response parsing error.

Interestingly, we did not observe any errors in the last two categories. Furthermore, only 1, 2, and 3 errors in categories 6, 8, and 7, respectively. The partition of the errors to the other 5 categories (see Figure 5), shows how much the models differ in the type of errors produced. Interestingly, the DeepSeek-Coder-V2 model rarely produces code that triggers exception or changes the input state and even typically passes the goal soundness test. Other models, especially the smaller ones, are more diverse in errors produced. Across all models, the majority of the errors account for the failed successor soundness and completeness tests.

**Bloopers** We noticed a few "bloopers", interesting phenomena that occur during AutoToS. We share these observations in a hope of shedding some light onto future understanding of LLM code generation for planning and search problems.

The first blooper occurs in the 5x5 Crossword for Llama3.1-70b. The representation of a Crossword instance includes vertical and horizontal clues which are lists of 5 words each. The model handles horizontal clues well by simply checking whether a word in row i is in the ith list in horizontal clues. For vertical clues, however, the model checks whether the word in column i is at position i among the clues for every column. The initial prompt from obtaining successor function clearly states that:

> ... horizontal_answers is a list where element i is a list of possible answers to clue in row i, and vertical_answers is a list where element i is a list of possible answers to clue in column i.

The second blooper occurs in the GPT-4o-mini, Llama3.1-70b, and even in Llama3.1-405b on the BlocksWorld domain. When generating successors for the unstack block from another block action, the models check if the block is clear, but never actually check whether the arm is empty. The resulting code, in cases when a block is already held, can generate a successor state in which the held block is overwritten with the one that is unstacked, and therefore disappears from the state. In some instances in the evaluation set the situation does not occur. In others, invalid solutions are produced and the accuracy score falls far below 100%. The AutoToS feedback in the next iterations often solves the problem.

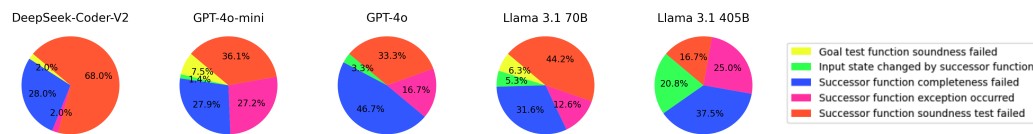

Figure 5: Partition of the errors by types in the generated code.

Another blooper occurs in Sokoban, when Llama3.1-70b generates the initial successor function and the goal test, and no partial soundness check is performed. The model generates a helper function *is_clear* that only checks whether the location on the grid is 0 or 2 (not a wall), disregarding whether any of the stones are currently at the location. This allows the player to move and push stones to the locations of other stones, resulting in the accuracy score of 0. Since the unit tests pass in this case, no additional iterations were performed. The partial soundness check would catch the error the first time a faulty state is generated (a state where multiple stones are at the same location or a player and a stone are at the same location). The prompt explicitly states what it means to be clear:

> The maze is defined by a grid of values 0,1, and 2, where 2 means it is a goal location for a stone, 1 means the cell is blocked, and either 0 or 2 means that the cell can be occupied. A cell is clear if it can be occupied but is not occupied by either the player or any stone.

Yet another blooper occurs in 24 Game with GPT-4o-mini and DeepSeek-CoderV2 when no partial soundness check is performed. When creating a new state out of the input state, two numbers are chosen to perform an arithmetic operation and in order to obtain the remaining numbers, the code selects the numbers from the state that are different from the two chosen numbers. Thus, in cases of duplicate numbers, the state size becomes more than one smaller than of their parent and on some instances the produced solutions would not be valid. The AutoToS completeness feedback eventually solves the problem in each of these cases.

## CONCLUSIONS AND FUTURE WORK

**Conclusions** We make the first step towards automating the process of generating code for the search components by leveraging debugging and exception handing with natural language, code feedback, and iterative reprompting. We demonstrate the performance of our approach, AutoToS, across various sized models and across representative search problems of various complexity. With just a few calls to the language model, we demonstrate that we can obtain the search components without any direct human in the loop feedback, ensuring soundness, completeness, accuracy, and nearly 100% accuracy across all models and all domains. We limit the human involvement to creating unit tests which therefore only requires problem domain understanding, and only limited or no coding skills.

**Limitations** We acknowledge inherent limitations of any such approach. First, the proposed procedure lacks convergence guarantees, as we have no theoretical way to ensure that the code quality improves from one iteration to another. Second, as with any other software testing, there is no guarantee of test coverage. Hence, the soundness and completeness cannot be guaranteed, even when all tests pass. Additionally, a limitation that can be relaxed in future work is that the benchmarks only capture a subset of planning problems that are fully observable and deterministic.

**Societal Impact** As with any technology, improving the planning abilities of LLMs can have positive as well as negative impacts. An example of positive impact is increased automated productivity, the flip side of which is loss of jobs. Awareness of pros and cons is crucial for policy generation.

**Future Work** For future work, we would like to further reduce human involvement in AutoToS. While existing work deals with generating unit tests (Pan et al., 2025) with the help of LLMs, it would be interesting to see if they could generate partial soundness tests, instead of relying on the user writing these for a specific domain. These partial soundness tests are related to the notion of invariants in planning (Alcázar & Torralba, 2015). It is worth exploring whether LLMs can help us derive such invariants. Further, it would be interesting to relax some of the assumptions, like full observability. Finally, seeing that smaller language models can achieve accuracy on par with the largest ones, begs the question of whether it would be possible to finetune an even smaller than the tested models and achieve similar or better accuracy.

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
