# OpenReview forum: "Automating Thought of Search: A Journey Towards Soundness and Completeness"
_ICLR.cc/2026/Conference — ICLR 2026 Conference Withdrawn Submission_

### Official Review · Reviewer_s5Fz · 2025-10-26

**Soundness:** 4
**Presentation:** 4
**Contribution:** 2
**Rating:** 4
**Confidence:** 4

**Summary:**

The work builds on previous method Thought of Search. In ToS, the LLM is prompted for code on goal-testing function an a successor-generation function. These funkction can be used via search methods BFS or DFS, etc.  While ToS achieves 100% accuracy, in previous work,  it needs a  human expert in the loops for evaluation. This is applied to planning, games etc where a tree of alternatives can be searched / explored to find a solution. The method is straight forward to implement.

**Strengths:**

1. It takes out the human in the loop for evaluation which makes it more usefull for practical application.
2. It allows for the use of smaller models since it splits to planning proplen into steps.
3. It is straigh forward to implement to replicated.
4. Good evaluation with a number of models and good number of problems.

**Weaknesses:**

1. It is expensive as each step needs a LLM call and this comes also with some latency.
2.  The paper claims 100% accuracy on tested datasets. This is great but comes with the downside that it does not allow access to the limits too. Benchmarks marks would be preferable that provide some space. Especially, blocksworld that would be possible by using more blocks or needs more plannings steps to solve.
3. The paper runs a best-of-5 as they have a validator so this seems to make it more expenstive but it can run in parallel, i belive.
4. Dependency on Test-Case Quality as knowledged by the authors ... also a limit for humans, we miss things too.

**Questions:**

Could you easily run this on task which are harder and to challenge this method and find problem cases? Where does it fail? One weak point was the test-case quality but if you make a problem really hard how far can you get? What problems occure and what do you think how to fix?

---

> ### Author Response · Authors · 2025-11-17
>
> Thank you for your constructive feedback and for raising these questions. We try to clarify below in the hopes of alleviating your concerns. We are very open to discussion!
>
>
> **W1**:
> AutoToS aims at generating a successor function and goal test code that is generic and works for all instances of a domain. The number of calls specified in Table 2 is therefore the total number of LLM calls needed per domain, not per instance in the domain. So, the number of LLM calls performed is actually very small.
> To clarify how small, the total number of calls to the language models across all the experiments, including 5 repetitions, is as follows: GPT-4o: 80,  GPT-4o-mini: 198, Llama3.1-405b: 76, Llama3.1-70b: 148, DeepSeek-CoderV2: 100.
> To compare, the famous Tree of Thought (ToT) [1] method performs hundreds of calls to the language model for each instance of each domain. Quoting the numbers from Table 1 of Katz et al, NeurIPS 2024, ToT would need to make up to a million calls for the same dataset we experiment with in this work, per repetition.
>
> **W2 & Q**:
> Our experiments aim at testing whether we were successful in generating the correct code for these search components. Therefore, there is no need here for larger instances, as the aim in this work is not to stress-test the search algorithms. In this work, we simply use a blind search algorithm like BFS or DFS with the LLM-generated code to test whether the code works for all the test cases. Scalability requires something beyond the correct search components, like heuristic functions. Existing work [2,3] already tackles this exact issue, under the assumption that the correct search components are available, and therefore our work is critical to the success of these approaches.
>
>
> [1] Shunyu Yao, Dian Yu, Jeffrey Zhao, Izhak Shafran, Tom Griffiths, Yuan Cao, and Karthik Narasimhan. Tree of Thoughts: Deliberate problem solving with large language models. In Proceedings of the Thirty-Seventh Annual Conference on Neural Information Processing Systems (NeurIPS 2023), 2023.
> [2] Alexander Tuisov, Yonatan Vernik, and Alexander Shleyfman. LLM-generated heuristics for AI Planning: Do we even need domain-independence anymore?, 2025. URL https://arxiv.org/abs/2501.18784.
> [3] Augusto B. Correa, Andre G. Pereira, and Jendrik Seipp. Classical planning with LLM-generated heuristics: Challenging the state of the art with python code, (NeurIPS 2025) 2025. URL https://arxiv.org/abs/2503.18809.

---

### Official Review · Reviewer_odni · 2025-10-31

**Soundness:** 2
**Presentation:** 3
**Contribution:** 2
**Rating:** 4
**Confidence:** 4

**Summary:**

The paper is an extension of prior work, Thought of Search (ToS), in which an LLM is used to generate Python code for a successor function and a goal test function which are then used with standard search algorithms like BFS or DFS to solve the given planning problems. While ToS required human in the loop to iteratively generate a sound successor function and a goal test function using LLMs, the paper aims to automate this step by guiding LLMs to generate a sound and complete successor function and a correct goal test function by using generic and domain specific unit tests for these functions as feedback to LLMs.

**Strengths:**

- The paper is easy to follow, and the contributions of the paper in the context of existing works are clearly communicated.
- The paper presents an important step towards automation of ToS enabling automated planning without requiring humans in the loop.
- The paper’s focus on the necessity of soundness and completeness in the context of planning (which are often overshadowed in works that tend to use LLMs directly as planners) is well-appreciated, as they are key to achieving good and reliable planning performance.
- The paper presents comprehensive experimental evaluation in different planning problems with varying complexity.

**Weaknesses:**

- The paper is incremental as it takes one specific step of the prior approach, and automates that step with an LLM-in-the-loop mechanism replacing the human-in-the-loop mechanism of prior approach. While this is an important extension, this limits the novelty and impacts of the paper to the broader research community.
- The claim about 100% accuracy feels underwhelming since it is based on success in any of the 5 trials (using an external validator). It would be interesting to see what average success rates look like for these experiments to get a better idea of the effectiveness of the proposed approach.

**Questions:**

- In line with my comment above about best-of-five trials when reporting accuracy, wouldn’t it be more meaningful to report average accuracy, since it is a more accurate reflection of the system’s performance when deployed in realistic settings without humans in the loop, as the paper aims to envision?
- In the discussion of results shown in Table 2, the authors claim that the number of LLM calls of their approach is comparable to that of ToS where human gives feedback, but the numbers in the table show that their approach consistently has higher number of calls compared to ToS. On what grounds are the authors making this claim? Are the authors considering feedback by humans in ToS as “pseudo” calls? This point should be clarified.

---

> ### Author Response · Authors · 2025-11-17
>
> Thank you for your constructive feedback and for raising these questions. We will integrate your suggestion into the next version of the paper, soon to be updated. We try to clarify below in the hopes of alleviating your concerns. We are very open to discussion!
>
> **W1. On the approach being incremental**:
>
> The current work is a step towards making the visionary approach of ToS into a practical method for deriving sound and complete solvers for real world planning problems. We cannot and should not only perform a shallow exploration of the space of scientific discoveries, we should dig deep, building on top of the existing knowledge. This work does precisely that.
>
> **W2 & Q1. On best-of-five**:
>
> The average accuracy (success rate) is depicted in Figure 3.
>
> In this work, we deal with planning problems for which sound validators exist. In such setting, we can run multiple times (in parallel or sequentially if time permits), testing if a solution is valid. We agree that in cases where no validators exist, the average behavior would be the right measure.
>
> To clarify, our best-of-five decision is not made per instance, but per domain, as our aim in the evaluation is to understand whether correct successor function and goal test code can be generated by AutoToS.
>
> **Q2**. On the comparability of the number of LLM calls:
> Table 2 shows the total number of calls a model makes in a domain, to solve all problems in that domain. To clarify, AutoToS makes the total of 10 calls to GPT-4o-mini to get the successor function and goal test code that are used to solve all 502 instances of blocksworld. If there were 5000 or 5M instances, the number of LLM calls would not change. In that sense, it is comparable to the 3.8 calls to GPT-4 that ToS is making with human feedback. For comparison, Tree of Thought (ToT) [1] will make around 200,000 LLM calls attempting to solve these 502 instances.
>
> [1] Shunyu Yao, Dian Yu, Jeffrey Zhao, Izhak Shafran, Tom Griffiths, Yuan Cao, and Karthik Narasimhan. Tree of Thoughts: Deliberate problem solving with large language models. In Proceedings of the Thirty-Seventh Annual Conference on Neural Information Processing Systems (NeurIPS 2023), 2023.

---

### Official Review · Reviewer_Fcu6 · 2025-10-31

**Soundness:** 2
**Presentation:** 2
**Contribution:** 1
**Rating:** 2
**Confidence:** 3

**Summary:**

The paper focuses on utilizing LLMs to solve planning problems that require search. Previous methods like ToS defined a search space with code, but required a human in the loop to inspect. The paper proposed AuthToS to achieve 100% accuracy on 5 synthetic tasks without human intervention.

**Strengths:**

The paper is generally written clearly.

This direction of reducing LLM dependence on human feedback for complex reasoning has potential.

**Weaknesses:**

I don’t see how this method can be really useful in more practical scenarios. Since the experiment setting is very toy, if you only focus on tasks with available `isgoal` and `succ` functions, you can actually replace the LLM call with very simple human-written functions, even simpler than the prompts you write.

Besides, I don't believe the 100% accuracy in the experiment section is meaningful. It is trivial to write a simple search method to achieve 100\% accuracy on all of them, as long as you have access to the two above functions.

There are no indices for different sections, which I believe does not conform to the ICLR template’s requirement.

**Questions:**

I don’t find any details in the paper on the human evaluation protocol details, since as the authors mentioned there’re human intervention in ToS method in Table 2. My guess is there’s some predefined functions as ground truth for ToS, but this is unclear from the text.

It is also better to include more baseline methods in Table 2 comparison to show your method’s practical value. For example, include the simplest search pipeline baseline.

---

> ### Author Response · Authors · 2025-11-17
>
> We appreciate your support for the direction of this work. There seems to be some confusion about the nature of this work, we try to clarify it in our response below.
> We are very open to a discussion, please let us know if you still have questions or comments.
>
> **On weakness points 1 & 2**:
> We completely agree with the reviewer that *if* isgoal and succ functions were available, one could run a BFS or DFS search to solve these problems.
>
> We would like to clarify that the isgoal and succ functions are *not* available in this work, only the natural language description of the domain and the unit tests. This work focuses precisely on generating these functions with the help of language models.
>
> Our experiments, therefore, are measuring the success in generating these functions correctly, and 100\% accuracy indicates that we are successful.
>
> Our evaluation is performed on a representative set of standard domains for search and planning problems.
>
> **On weakness point 3**:
> We use ICLR Latex style. We can change `\setcounter{secnumdepth}{0}` to `1`, if the reviewer finds section numbers useful. We did not reference the sections and therefore did not find it necessary to use section numbers.
>
> **Q1**. We did not redo the experiments on ToS, and simply used the numbers reported in that work.
>
> **Q2**. Could you please clarify what do you mean by "the simplest search pipeline baseline"?

---

### Official Review · Reviewer_F3y3 · 2025-10-31

**Soundness:** 3
**Presentation:** 2
**Contribution:** 3
**Rating:** 4
**Confidence:** 4

**Summary:**

This paper extends the so-caled thought-of-search (ToS) method which basically uses code to structure the search for solutions. Like a generic A* search algorithm this defines a successor function and a goal test function which can then be compared to any other search approach executed on the same environment.

**Strengths:**

- The main contribution, according to the authors, is that they can leverage all the existing approaches in software engineering and LLM-based code generation to create an automated feedback mechanism thus avoiding need for human intervention. (But it looks like the human still has to generate some unit tests, unless you are using formal specs of the environment?)
- They show that this approach leads to 100% accuracy (just like the manually intervened ToS models).
- The error analysis is good showing distinctions across the different models.

**Weaknesses:**

- While I like the paper's approach this seems like a very minor contribution and primarily relies on test-based checks to achieve full 100% automation. The choice of problems might have been restricted to those where this is possible. I am not sure how this can be used as a general reasoning mechanism for LLM applications.
- While I commended the error analysis above, this also feels like an opportunity to use this to see what needs to be done so you can get uniform performance across models and domains.

**Questions:**

- Please address the issues raised in the weaknesses section.

---

> ### Author Response · Authors · 2025-11-17
>
> Thank you for your support, your constructive feedback and for raising these questions. We try to clarify below in the hopes of alleviating your concerns. We are very open to discussion!
>
> **W1**
> In this work, we are dealing with sequential decision making/planning problems, not with general reasoning problems. We experiment with a variety of representative standard search problems. Search-based methods are a natural choice for these problems.
> Being able to reliably solve such problems is of immense importance, as these problems span real world applications from logistics to cybersecurity to program analysis to workflow orchestration and much more.
>
> **W2**
> Indeed, our error analysis can serve as a guidance for training data generation, good point! We do not see it as a weakness of this work that we do not deal with training the language models. It should be a separate research question and we are happy to see that the reviewer feels that our error analysis can contribute to that research question.

---

### Official Review · Reviewer_MK1H · 2025-11-03

**Soundness:** 3
**Presentation:** 2
**Contribution:** 2
**Rating:** 4
**Confidence:** 3

**Summary:**

The paper introduces an adaption of the Thought of Search (ToS) framework: AutoTos, a system that seeks to sideline humans' involvement in the feedback loop for generated coding snippets. The automation relies on domain-specific and generic unit tests to iteratively provide feedback to LLMs on their code outputs, aiming for code that is sound and complete. Experiments are conducted across five benchmark domains and with multiple LLMs.

**Strengths:**

- The paper clearly motivates the need to automate the iterative feedback and exception handling process within the LLM-driven ToS paradigm, effectively shifting the burden from continuous human involvement to the specification of unit tests by humans.
- Evaluation is conducted across multiple domains of varying complexity, including classic (i.e. with BlocksWorld and Sokoban), logic (i.e. with PrOntoQA), and mini-crosswords, as well as with various LLMs of different sizes.
- AutoToS often achieves 100% accuracy and does so with a manageable number of LLM calls, usually on par with the original ToS approach involving human experts, especially when using larger LLMs.

**Weaknesses:**

- Even though much of the process is automated, the framework assumes either existing or easily generated unit tests. Consequently, human involvement is not eliminated but rather shifted: from interacting directly with the LLM to designing or generating appropriate unit tests.
- The authors use examples from the 24 Game without providing sufficient explanations of the game itself. Including a brief description of the game in the Background section, ideally expanding the existing explanation on lines $275-277$, would help readers unfamiliar with the game better understand its rules and how states transition through arithmetic operations, thereby clarifying the relevance of the successor states used in the examples.
- The framework relies on soundness and completeness as feedback properties. While the experimental results show the benefits they bring, the paper itself does not include any comparison with other types of feedback or other feedback-guided approaches.

**Questions:**

- Could you please verify the equation on line 112? Shouldn't it be $f(s_i, a_i) = s_{i+1}$?
- What does $T$ represent on line 115?
- Providing more details on the limitations and challenges of generating the unit tests, as well as possible coverage metrics for those tests, would help clarify the full extent of the remaining human effort.

---

> ### Author Response · Authors · 2025-11-17
>
> Thank you for your constructive feedback and for raising these questions. We will integrate your suggestion into the next version of the paper, soon to be updated. We try to clarify below in the hopes of alleviating your concerns. We are very open to discussion!
>
>
> **W1**: You are absolutely correct that AutoToS is not a fully automated process. The human involvement is shifted to a different stage of the process. Importantly, it is shifted from humans that need to be both domain experts as well as skilled software engineers, to be able to debug the generated code on the fly and feedback the language model on it to humans that are domain experts, with minimal to none software engineering skills. We strongly believe that this is a significant step in the right direction.
>
> **W3**: We compare to no feedback (Figure 3, "first") and to not caring for completeness (Figure 3, "sound"). We indeed do not experiment with different types of feedback, finding that CoT-style feedback works well. The performance of other feedback-guided approaches popular in the literature, such as  ToT/GoT/AoT/LATS/RAP on these domains was already discussed in the previous work, e.g., by ToS (Katz et al., NeurIPS 2024).
>
> **Q1**: The equation on line 112 is correct as stated. As common in the planning community, we start the action index from 1 (please see the trace in line 111).
>
> **Q2**: T and F on line 115 stand for True and False. We will clarify in text.
>
> **Q3**: In the domains we experiment with in this work, the unit tests are very simple. For 24 game, as an example, an example non-goal state [24,1] is provided as a unit test (testing that the return value for this state is false), an example set of states and their successors is provided, to test that all the provided successors are produced by the generated code.
> An (optional) partial soundness check would simply test that all the successors generated by the generated code have size exactly one smaller than the input state - meaning that the successor function does not skip steps. Please see Appendix A for the complete list of unit tests for the experimented domains. In many practical domains such information is already readily available, e.g., as historical traces.

---

### Note · Authors · 2025-11-30

I have read and agree with the venue's withdrawal policy on behalf of myself and my co-authors.